# Language as an Abstraction
# for Hierarchical Deep Reinforcement Learning

**Yiding Jiang**\*, **Shixiang Gu, Kevin Murphy, Chelsea Finn**
Google Research
{ydjiang,shanegu,kpmurphy,chelseaf}@google.com

## Abstract

Solving complex, temporally-extended tasks is a long-standing problem in reinforcement learning (RL). We hypothesize that one critical element of solving such problems is the notion of *compositionality*. With the ability to learn concepts and sub-skills that can be composed to solve longer tasks, i.e. hierarchical RL, we can acquire temporally-extended behaviors. However, acquiring effective yet general abstractions for hierarchical RL is remarkably challenging. In this paper, we propose to use language as the abstraction, as it provides unique compositional structure, enabling fast learning and combinatorial generalization, while retaining tremendous flexibility, making it suitable for a variety of problems. Our approach learns an instruction-following low-level policy and a high-level policy that can reuse abstractions across tasks, in essence, permitting agents to reason using structured language. To study compositional task learning, we introduce an open-source object interaction environment built using the MuJoCo physics engine and the CLEVR engine. We find that, using our approach, agents can learn to solve to diverse, temporally-extended tasks such as object sorting and multi-object rearrangement, including from raw pixel observations. Our analysis reveals that the compositional nature of language is critical for learning diverse sub-skills and systematically generalizing to new sub-skills in comparison to non-compositional abstractions that use the same supervision.[2]

## 1 Introduction

Deep reinforcement learning offers a promising framework for enabling agents to autonomously acquire complex skills, and has demonstrated impressive performance on continuous control problems [35, 56] and games such as Atari [41] and Go [59]. However, the ability to learn a variety of compositional, long-horizon skills while generalizing to novel concepts remain an open challenge. Long-horizon tasks demand sophisticated exploration strategies and structured reasoning, while generalization requires suitable representations. In this work, we consider the question: how can we leverage the compositional structure of *language* for enabling agents to perform long-horizon tasks and systematically generalize to new goals?

To do so, we build upon the framework of hierarchical reinforcement learning (HRL), which offers a potential solution for learning long-horizon tasks by training a hierarchy of policies. However, the abstraction between these policies is critical for good performance. Hard-coded abstractions often lack modeling flexibility and are task-specific [63, 33, 26, 47], while learned abstractions often find degenerate solutions without careful tuning [5, 24]. One possible solution is to have the higher-level policy generate a sub-goal state and have the low-level policy try to reach that goal state [42, 36]. However, using goal states still lacks some degree of flexibility (e.g. in comparison



(a) Goal is $g_0$: "*There is a* **red** *ball; are there any matte* **cyan** *sphere* **right** *of it?*". Currently $\Psi(s_t, g_0) = 0$

(b) Agent performs actions and interacts with the environment and tries to satisfy goal.

(c) Resulting state $s_{t+1}$ does not satisfy $g_0$, so relabel goal to $g'$: " *There is a* **green** *sphere; are there any rubber* **cyan** *balls* **behind** *it?*" so $\Psi(s_{t+1}, g') = 1$

Figure 1: The environment and some instructions that we consider in this work, along with an illustration of hindsight instruction relabeling (HIR), which we use to enable the agent to learn from many different language goals at once (Details in Section 4.2).

to goal regions or attributes), is challenging to scale to visual observations naively, and does not generalize systematically to new goals. In contrast to these prior approaches, language is a flexible representation for transferring a variety of ideas and intentions with minimal assumptions about the problem setting; its compositional nature makes it a powerful abstraction for representing combinatorial concepts and for transferring knowledge [22].

In this work, we propose to use language as the interface between high- and low-level policies in hierarchical RL. With a low-level policy that follows language instructions (Figure 1), the high-level policy can produce actions in the space of language, yielding a number of appealing benefits. First, the low-level policy can be re-used for different high-level objectives without retraining. Second, the high-level policies are human-interpretable as the actions correspond to language instructions, making it easier to recognize and diagnose failures. Third, language abstractions can be viewed as a strict generalization of goal states, as an instruction can represent a *region of states* that satisfy some abstract criteria, rather than the entirety of an individual goal state. Finally, studies have also suggested that humans use language as an abstraction for reasoning and planning [21, 49]. In fact, the majority of knowledge learning and skill acquisition we do throughout our life is through languages.

While language is an appealing choice as the abstraction for hierarchical RL, training a low-level policy to follow language instructions is highly non-trivial [20, 6] as it involves learning from binary rewards that indicate completion of the instruction. To address this problem, we generalize prior work on goal relabeling to the space of language instructions (which instead operate on regions of state space, rather than a single state), allowing the agent to learn from many language instructions at once.

To empirically study the role of language abstractions for long-horizon tasks, we introduce a new environment inspired by the CLEVR engine [28] that consists of procedurally-generated scenes of objects that are paired with programatically-generated language descriptions. The low-level policy's objective is to manipulate the objects in the scene such that a description or statement is satisfied by the arrangement of objects in the scene. We find that our approach is able to learn a variety of vision-based long-horizon manipulation tasks such as object reconfiguration and sorting, while outperforming state-of-the-art RL and hierarchical RL approaches. Further, our experimental analysis finds that HRL with non-compositional abstractions struggles to learn the tasks, even when the non-compositional abstraction is derived from language instructions themselves, demonstrating the critical role of compositionality in learning. Lastly, we find that our instruction-following agent is able to generalize to instructions that are systematically different from those seen during training.

In summary, the main contribution of our work is three-fold:

1. a framework for using language abstractions in HRL, with which we find that the structure and flexibility of language enables agents to solve challenging long-horizon control problems

2. an open-source continuous control environment for studying compositional, long-horizon tasks, integrated with language instructions inspired by the CLEVR engine [28]

3. empirical analysis that studies the role of compositionality in learning long-horizon tasks and achieving systematic generalization

## 2 Related Work

Designing, discovering and learning meaningful and effective abstractions of MDPs has been studied extensively in hierarchical reinforcement learning (HRL) [15, 46, 63, 16, 5]. Classically, the work

on HRL has focused on learning only the high-level policy given a set of hand-engineered low-level policies [60, 38, 9], or more generically *options* policies with flexible termination conditions [63, 52].

Recent HRL works have begun to tackle more difficult control domains with both large state spaces and long planning horizons [26, 34, 65, 17, 42, 43]. These works can typically be categorized into two approaches. The first aims to learn effective low-level policies end-to-end directly from final task rewards with minimal human engineering, such as through the option-critic architecture [5, 24] or multi-task or meta learning [18, 58]. While appealing in theory, this end-to-end approach relies solely on final task rewards and is shown to scale poorly to complex domains [5, 42], unless distributions of tasks are carefully designed [18]. The second approach instead augments the low-level learning with auxiliary rewards that can bring better inductive bias. These rewards include mutual information-based diversity rewards [13, 17], hand-crafted rewards based on domain knowledge [33, 26, 34, 65], and goal-oriented rewards [15, 55, 4, 69, 42, 43]. Goal-oriented rewards have been shown to balance sufficient inductive bias for effective learning with minimal domain-specific engineering, and achieve performance gains on a range of domains [69, 42, 43]. Our work is a generalization on these lines of work, representing goal *regions* using language instructions, rather than individual goal *states*. Here, *region* refers to the sets of states (possibly disjoint and far away from each other) that satisfy more abstract criteria (e.g. *"red ball in front of blue cube"* can be satisfied by infinitely many states that are drastically different from each other in the pixel space) rather than a simple $\epsilon$-ball around a single goal state that is only there for creating a reachable non-zero measure goal. Further, our experiments demonstrate significant empirical gains over these prior approaches.

Since our low-level policy training is related to goal-conditioned HRL, we can benefit from algorithmic advances in multi-goal reinforcement learning [29, 55, 4, 50]. In particular, we extend the recently popularized goal relabeling strategy [29, 4] to instructions, allowing us to relabel based on achieving a language statement that describes a *region* of state space, rather than relabeling based on reaching an individual state.

Lastly, there are a number of prior works that study how language can guide or improve reinforcement learning [37, 19, 30, 6, 20, 12, 10]. While prior work has made use of language-based sub-goal policies in hierarchical RL [57, 14], the instruction representations used lack the needed diversity to benefit from the compositionality of language over one-hot goal representations. In a concurrent work, Wu et al. [70] show that language can help with learning difficult tasks where more naive goal representations lead to poor performance, even with hindsight goal relabeling. While we are also interested in using language to improve learning of challenging tasks, we focus on the use of language in the context of hierarchical RL, demonstrating that language can be further used to compose complex objectives for the agent. Andreas et al. [3] leverage language descriptions to rapidly adapt to unseen environments through structured policy search in the space of language; each environment is described by one sentence. In contrast, we show that a high-level policy can effectively leverage the combinatorially many sub-policies induced by language by generating a sequence of instructions for the low-level agent. Further, we use language not only for adaptation but also for learning the lower level control primitives, without the need for imitation learning from an expert. Another line of work focuses on RL for textual adventure games where the state is represented as language descriptions and the actions are either textual actions available at each state [25] or all possible actions [45] (even though not every action is applicable to all states). In general, these works look at text-based games with discrete 1-bit actions, while we consider continuous actions in physics-based environments. One may view the latter as a high-level policy with oracular low-level policies that are specific to each state; the discrete nature of these games entails limited complexity of interactions with the environment.

## 3    Preliminaries

**Standard reinforcement learning.** The typical RL problem considers a *Markov decision process* (MDP) defined by the tuple $(\mathcal{S}, \mathcal{A}, T, R, \gamma)$ where $\mathcal{S}$ where $\mathcal{S}$ is the state space, $\mathcal{A}$ is the action space, the unknown transition probability $T : \mathcal{S} \times \mathcal{A} \times \mathcal{S} \to [0, \infty)$ represents the probability density of reaching $s_{t+1} \in \mathcal{S}$ from $\mathbf{s}_t \in \mathcal{S}$ by taking the action $\mathbf{a} \in \mathcal{A}$, $\gamma \in [0, 1)$ is the discount factor, and the bounded real-valued function $R : \mathcal{S} \times \mathcal{A} \to [r_{\min}, r_{\max}]$ represents the reward of each transition. We further denote $\rho_\pi(\mathbf{s}_t)$ and $\rho_\pi(\mathbf{s}_t, \mathbf{a}_t)$ as the state marginal and the state-action marginal of the trajectory induced by policy $\pi(\mathbf{a}_t|\mathbf{s}_t)$. The objective of reinforcement learning is to find a policy $\pi(\mathbf{a}_t|\mathbf{s}_t)$ such that the expected discounted future reward $\sum_t \mathbb{E}_{(\mathbf{s}_t, \mathbf{a}_t) \sim \rho_\pi}[\gamma^t R(\mathbf{s}_t, \mathbf{a}_t)]$ is maximized.

**Goal conditioned reinforcement learning.** In goal-conditioned RL, we work with an *Augmented Markov Decision Process*, which is defined by the tuple $(\mathcal{S}, \mathcal{G}, \mathcal{A}, T, R, \gamma)$. Most elements represent the same quantities as a standard MDP. The additional tuple element $\mathcal{G}$ is the space of all possible goals, and the reward function $R : \mathcal{S} \times \mathcal{A} \times \mathcal{G} \to [r_{\min}, r_{\max}]$ represents the reward of each transition *under a given goal*. Similarly, the policy $\pi(\boldsymbol{a}_t|\boldsymbol{s}_t, \boldsymbol{g})$ is now conditioned on $\boldsymbol{g}$. Finally, $p_g(\boldsymbol{g})$ represents a distribution over $\mathcal{G}$. The objective of goal directed reinforcement learning is to find a policy $\pi(\boldsymbol{a}_t|\boldsymbol{s}_t, \boldsymbol{g})$ such that the expected discounted future reward $\sum_t \mathbb{E}_{\boldsymbol{g} \sim p_g, (\boldsymbol{s}_t, \boldsymbol{a}_t) \sim \rho_\pi}[\gamma^t R(\boldsymbol{s}_t, \boldsymbol{a}_t, \boldsymbol{g})]$ is maximized. While this objective can be expressed with a standard MDP by augmenting the state vector with a goal vector, the policy does not change the goal; the explicit distinction between goal and state facilitates discussion later.

**Q-learning.** Q-learning is a large class of off-policy reinforcement learning algorithms that focuses on learning the Q-function, $Q^*(\boldsymbol{s}_t, \boldsymbol{a}_t)$, which represents the expected total discounted reward that can be obtained after taking action $\boldsymbol{a}_t$ in state $\boldsymbol{s}_t$ assuming the agent acts optimally thereafter. It can be recursively defined as:

$$Q^*(\boldsymbol{s}_t, \boldsymbol{a}_t) = \mathbb{E}_{\boldsymbol{s}_{t+1}}[R(\boldsymbol{s}_t, \boldsymbol{a}_t) + \gamma \max_{\boldsymbol{a} \in \mathcal{A}}(Q^*(\boldsymbol{s}_{t+1}, \boldsymbol{a}))] \tag{1}$$

The optimal policy learned can be recovered through $\pi^*(\boldsymbol{a}_t|\boldsymbol{s}_t) = \delta(\boldsymbol{a}_t = \arg \max_{\boldsymbol{a} \in \mathcal{A}} Q^*(\boldsymbol{s}_t, \boldsymbol{a}))$. In high-dimensional spaces, the Q-function is usually represented with function approximators and fit using transition tuples, $(\boldsymbol{s}_t, \boldsymbol{a}_t, \boldsymbol{s}_{t+1}, r_t)$, which are stored in a *replay buffer* [41].

**Hindsight experience replay (HER).** HER [4] is a data augmentation technique for off-policy goal conditioned reinforcement learning. For simplicity, assume that the goal is specified in the state space directly. A trajectory can be transformed into a sequence of goal augmented transition tuples $(\boldsymbol{s}_t, \boldsymbol{a}_t, \boldsymbol{s}_g, \boldsymbol{s}_{t+1}, r_t)$. We can relabel each tuple's $\boldsymbol{s}_g$ with $\boldsymbol{s}_{t+1}$ or other future states visited in the trajectory and adjust $r_t$ to be the appropriate value. This makes the otherwise sparse reward signal much denser. This technique can also been seen as generating an implicit curriculum of increasing difficulty for the agent as it learns to interact with the environment more effectively.

# 4 Hierarchical Reinforcement Learning with Language Abstractions

In this section, we present our framework for training a 2-layer hierarchical policy with compositional *language* as the abstraction between the high-level policy and the low-level policy. We open the exposition with formalizing the problem of solving temporally extended task with language, including our assumptions regarding the availability of supervision. We will then discuss how we can efficiently train the low-level policy, $\pi_l(\boldsymbol{a}|\boldsymbol{s}_t, \boldsymbol{g})$ conditioned on language instructions $\boldsymbol{g}$ in Section 4.2, and how a high-level policy, $\pi_h(\boldsymbol{g}|\boldsymbol{s}_t)$, can be trained using such a low-level policy in Section 4.3. We refer to this framework as *Hierarchical Abstraction with Language* (HAL, Figure 2, Appendix C.1).

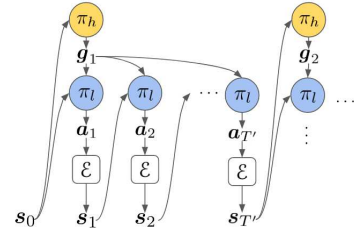

Figure 2: HAL: The high-level policy $\pi_h$ produces language instructions $\boldsymbol{g}$ for the low level policy $\pi_l$.

## 4.1 Problem statement

We are interested in learning temporally-extended tasks by leveraging the compositionality of language. Thus, in addition to the standard reinforcement learning assumptions laid out in Section 3, we also need some form of grounded language supervision in the environemnt $\mathcal{E}$ during training. To this end, we also assume the access to a conditional density $\omega(\boldsymbol{g}|\boldsymbol{s})$ that maps observation $\boldsymbol{s}$ to a distribution of language statements $\boldsymbol{g} \in \mathcal{G}$ that describes $\boldsymbol{s}$. This distribution can take the form of a supervised image captioning model, a human supervisor, or a functional program that is executed on $\boldsymbol{s}_t$ similar to CLEVR. Further, we define $\Omega(\boldsymbol{s}_t)$ to be the support of $\omega(\boldsymbol{g}|\boldsymbol{s}_t)$. Moreover, we assume the access to a function $\Psi$ that maps a state and an instruction to a single Boolean value which indicates whether the instruction is satisfied or not by the $\boldsymbol{s}$, i.e. $\Psi : \mathcal{S} \times \mathcal{G} \to \{0, 1\}$. Once again, $\Psi$ can be a VQA model, a human supervisor or a program. Note that any goal specified in the state space can be easily expressed by a Boolean function of this form by checking if two states are close to each other up to some threshold parameter. $\Psi$ can effectively act as the reward for the low-level policy.

An example for the high-level tasks is arranging objects in the scene according to a specific spatial relationship. This can be putting the object in specific arrangement or ordering the object according

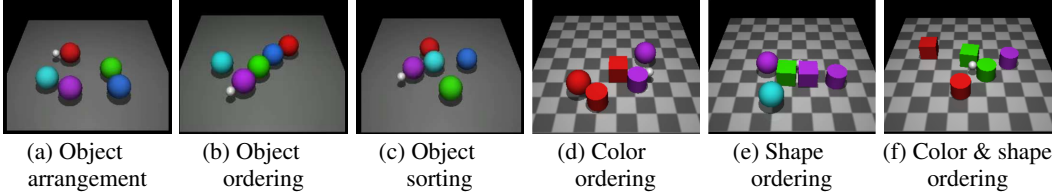

| (a) Object arrangement | (b) Object ordering | (c) Object sorting | (d) Color ordering | (e) Shape ordering | (f) Color & shape ordering |

Figure 3: Sample goal states for the high-level tasks in the standard (a-c) and diverse (d-f) environments. The high-level policy only receives reward if all constraints are satisfied. The global location of the objects may vary.

the the colors (Figure 3) by pushing the objects around (Figure 1). Details of these high-level tasks are described in Section 5. These tasks are complex but can be naturally decomposed into smaller sub-tasks, giving rise to a naturally defined hierarchy, making it an ideal testbed for HRL algorithms. Problems of similar nature including organizing a cluttered table top or stacking LEGO blocks to construct structures such as a castle or a bridge. We train the low-level policy $\pi_l(\boldsymbol{a}|\boldsymbol{s}, \boldsymbol{g})$ to solve an augmented MDP described Section 3. For simplicity, we assume that $\Omega$'s output is uniform over $\mathcal{G}$. The low-level policy receives supervision from $\Omega$ and $\Psi$ by completing instructions. The high-level policy $\pi_h(\boldsymbol{g}|\boldsymbol{s})$ is trained to solve a standard MDP whose state space is the same $\mathcal{S}$ as the low-level policy, and the action space is $\mathcal{G}$. In this case, the high-level policy's supervision comes from the reward function of the environment which may be highly sparse.

We separately train the high-level policy and low-level policy, so the low-level policy is agnostic to the high-level policy. Since the policies share the same $\mathcal{G}$, the low-level policy can be reused for different high-level policies (Appendix C.3). Jointly fine-tuning the low-level policy with a specific high-level policy is certainly a possible direction for future work (Appendix C.1).

## 4.2 Training a language-conditioned low-level policy

To train a goal conditioned low-level policy, we need to define a suitable reward function for training such a policy and a mechanism for sampling language instructions. A straightforward way to represent the reward for the low-level policy would be $R(\boldsymbol{s}_t, \boldsymbol{a}_t, \boldsymbol{s}_{t+1}, \boldsymbol{g}) = \Psi(\boldsymbol{s}_{t+1}, \boldsymbol{g})$ or, to ensure that $\boldsymbol{a}_t$ is inducing the reward:

$$R(\boldsymbol{s}_t, \boldsymbol{a}_t, \boldsymbol{s}_{t+1}, \boldsymbol{g}) = \begin{cases} 0 & \text{if } \Psi(\boldsymbol{s}_{t+1}, \boldsymbol{g}) = 0 \\ \Psi(\boldsymbol{s}_{t+1}, \boldsymbol{g}) \oplus \Psi(\boldsymbol{s}_t, \boldsymbol{g}) & \text{if } \Psi(\boldsymbol{s}_{t+1}, \boldsymbol{g}) = 1 \end{cases}$$

However, optimizing with this reward directly is difficult because the reward signal is only non-zero when the goal is achieved. Unlike prior work (e.g. HER [4]), which uses a state vector or a task-relevant part of the state vector as the goal, it is difficult to define meaningful distance metrics in the space of language statements [8, 53, 61], and, consequently, difficult to make the reward signal smooth by assigning partial credit (unlike, e.g., the $\ell_p$ norm of the difference between 2 states). To overcome these difficulties, we propose a trajectory relabeling technique for language instructions: Instead of relabeling the trajectory with states reached, we relabel states in the the trajectory $\tau$ with the elements of $\Omega(\boldsymbol{s}_t)$ as the goal instruction using a relabeling strategy $\mathcal{S}$. We refer to this procedure as *hindsight instruction relabeling* (HIR). The details of $\mathcal{S}$ is located in Algorithm 4 in Appendix C.4. Pseudocode for the method can be found in Algorithm 2 in Appendix C.2 and an illustration of the process can be found in Figure 1.

The proposed relabeling scheme, HIR, is reminiscent of HER [4]. In HER, the goal is often the state or a simple function of the state, such as a masked representation. However, with high-dimensional observation spaces such as images, there is excessive information in the state that is irrelevant to the goal, while task-relevant information is not readily accessible. While one can use HER with generative models of images [54, 51, 44], the resulting representation of the state may not effectively capture the relevant aspects of the desired task. Language can be viewed as an alternative, highly-compressed representation of the state that explicitly exposes the task structure, e.g. the relation between objects. Thus, we can readily apply HIR to settings with image observations.

## 4.3 Acting in language with the high-level policy

We aim to learn a high-level policy for long-horizon tasks that can explore and act in the space of language by providing instructions $\boldsymbol{g}$ to the low-level policy $\pi_l(\boldsymbol{a}|\boldsymbol{s}_t, \boldsymbol{g})$. The use of language abstractions through $\boldsymbol{g}$ allows the high-level policy to structurally explore with actions that are semantically meaningful and span multiple low-level actions.

In principle, the high-level policy, $\pi_h(\boldsymbol{g}|\boldsymbol{s})$, can be trained with any reinforcement learning algorithm, given a suitable way to generate sentences for the goals. However, generating coherent sequences of discrete tokens is difficult, particularly when combined with existing reinforcement learning algorithms. We explore how we might incorporate a language model into the high level policy in Appendix A, which shows promising preliminary results but also significant challenges. Fortunately, while the size of the instruction space $\mathcal{G}$ scales exponentially with the size of the vocabulary, the elements of $\mathcal{G}$ are naturally structured and redundant – many elements correspond to effectively the same underlying instruction with different synonyms or grammar. While the low-level policy understands all the different instructions, in many cases, the high-level policy only needs to generate instruction from a much smaller subset of $\mathcal{G}$ to direct the low-level policy. We denote such subsets of $\mathcal{G}$ as $\mathcal{I}$.

If $\mathcal{I}$ is relatively small, the problem can be recast as a discrete-action RL problem, where one action choice corresponds to an instruction, and can be solved with algorithms such as DQN [41]. We adopt this simple approach in this work. As the instruction often represents a sequence of low-level actions, we take $T'$ actions with the low-level policy for every high-level instruction. $T'$ can be a fixed number of steps, or computed dynamically by a terminal policy learned by the low-level policy like the option framework. We found that simply using a fixed $T'$ was sufficient in our experiments.

## 5 The Environment and Implementation

**Environment.** To empirically study how compositional languages can aid in long-horizon reasoning and generalization, we need an environment that will test the agent's ability to do so. While prior works have studied the use of language in navigation [1], instruction following in a grid-world [10], and compositionality in question-answering, we aim to develop a *physical* simulation environment where the agent must interact with and change the environment in order to accomplish long-horizon, compositional tasks. These criteria are particularly appealing for robotic applications, and, to the best of our knowledge, none of the existing benchmarks simultaneously fulfills all of them. To this end, we developed a new environment using the MuJoCo physics engine [66] and the CLEVR language engine, that tests an agents ability to manipulate and rearrange objects of various shapes and colors. To succeed in this environment, the agent must be able to handle varying number of objects with diverse visual and physical properties. Two versions of the environment of varying complexity are illustrated in Figures 3 and 1 and further details are in Appendix B.1.

**High-level tasks.** We evaluate our framework 6 challenging temporally-extended tasks across two environments, all illustrated in Figure 3: (a) *object arrangement*: manipulate objects such that 10 pair-wise constraints are satisfied, (b) *object ordering*: order objects by color, (c) *object sorting*: arrange 4 objects around a central object, and in a more diverse environment (d) *color ordering*: order objects by color irrespective of shape, (e) *shape ordering*: order objects by shape irrespective of color, and (f) *color & shape ordering*: order objects by both shape and color. In all cases, the agent receives a binary reward only if all constraints are satisfied. Consequently, this makes obtaining meaningful signal in these tasks extremely challenging as only a very small number of action sequences will yield non-zero signal. For more details, see Appendix B.2.

**Action and observation parameterization.** The state-based observation is $\boldsymbol{s} \in \mathbb{R}^{10}$ that represents the location of each object and $|\mathcal{A}| = 40$ which corresponds to picking an object and pushing it in one of the eight cardinal directions. The image-based observation is $\boldsymbol{s} \in \mathbb{R}^{64 \times 64 \times 3}$ which is the rendering of the scene and $|\mathcal{A}| = 800$ which corresponds to picking a location in a $10 \times 10$ grid and pushing in one of the eight cardinal directions. For more details, see Appendix B.1.

**Policy parameterization.** The low-level policy encodes the instruction with a GRU and feeds the result, along with the state, into a neural network that predicts the Q-value of each action. The high-level policy is also a neural network Q-function. Both use Double DQN [68] for training. The high-level policy uses sets of 80 and 240 instructions as the action space in the standard and diverse environments respectively, a set that sufficiently covers relationships between objects. We roll out the low-level policy for $T' = 5$ steps for every high-level instruction. For details, see Appendix B.3.

## 6 Experiments

To evaluate our framework, and study the role of compositionality in RL in general, we design the experiments to answer the following questions: **(1)** as a representation, how does language compare

to alternative representations, such as those that are not explicitly compositional? **(2)** How well does the framework scale with the diversity of instruction and dimensionality of the state (e.g. vision-based observation)? **(3)** Can the policy generalize in systematic ways by leveraging the structure of language? **(4)** Overall, how does our approach compare to state-of-the-art hierarchical RL approaches, along with learning flat, homogeneous policies?

To answer these questions, in Section 6.1, we first evaluate and analyze training of effective low-level policies, which are critical for effective learning of long-horizon tasks. Then, in Section 6.2, we evaluate the full HAL method on challenging temporally-extended tasks. Finally, we apply our method to the Crafting environment from Andreas et al. [2] to showcase the generality of our framework. For details on the experimental set-up and analysis, see Appendix D and E.

## 6.1 Low-level Policy

**Role of compositionality and relabeling.** We start by evaluating the fidelity of the low-level instruction-following policy, in isolation, with a variety of representations for the instruction. For these experiments, we use state-based observations. We start with a set of 600 instructions, which we paraphrase and substitute synonyms to obtain more than 10,000 total instructions which allows us to answer the first part of **(2)**. We evaluate the performance all low-level policies by the average number of instructions it can successfully achieve each episode (100 steps), measured over 100 episodes. To answer **(1)**, and evaluate the importance of compositionality, we compare against:

- a **one-hot encoded representation** of instructions where each instruction has its own row in a real-valued embedding matrix which uses the same instruction relabeling (see Appendix D.1)
- a **non-compositional latent variable representation** with identical information content. We train an sequence auto-encoder on sentences, which achieves 0 reconstruction error and is hence a *lossless* non-compositional representation of the instructions (see Appendix D.2)
- a **bag-of-words (BOW) representation** of instructions (see Appendix D.3)

In the first comparison, we observe that while one-hot encoded representation works on-par with or better than the language in the regime where the number of instructions is small, its performance quickly deteriorates as the number of instruction increases (Fig.4, middle). On the other hand, language representation of instruction can leverage the structures shared by different instructions and does not suffer from increasing number of instructions (Fig.4, right blue); in fact, an *improvement* in performance is observed. This suggests, perhaps unsurprisingly, that one-hot representations and state-based relabeling scale poorly to large numbers of instructions, *even when the underlying number of instructions does not change*, while, with instruction relabeling (HIR), the policy acquires better, more successful representations as the number of instructions increases.

In the second comparison, we observe that the agent is unable to make meaningful progress with this representation despite receiving identical amount of supervision as language. This indicates that the compositionality of language is critical for effective learning. Finally, we find that relabeling is critical for good performance, since without it (no HIR), the reward signal is significantly sparser.

Finally, in the comparison to a bag-of-words representation (BOW), we observe that, while the BOW agent's return increases at a faster rate than that of the language agent at the beginning of training – likely due to the difficulty of optimizing recurrent neural network in language agent – the language agent achieves significantly better final performance. On the other hand, the performance of the BOW agent plateaus at around 8 instructions per episode. This is expected as BOW does not consider the *sequential nature* of an instruction, which is important for correctly executing an instruction.

**Vision-based observations.** To answer the second part of **(2)**, we extend our framework to pixel observations. The agent reaches the same performance as the state-based model, albeit requiring longer convergence time with the same hyper-parameters. On the other hand, the one-hot representation reaches much worse relative performance with the same amount of experience (Fig.4, right).

**Visual generalization.** One of the most appealing aspects of language is the promise of *combinatorial generalization* [7] which allows for *extrapolation* rather than simple *interpolation* over the training set. To evaluate this (i.e. **(3)**), we design the training and test instruction sets that are systematically distinct. We evaluate the agent's ability to perform such generalization by splitting the 600 instruction sets through the following procedure: **(i) standard**: random 70/30 split of the instruction set; **(ii) systematic**: the training set only consists of instructions that do not contain the words *red* in the first half of the instructions and the test set contains only those that have *red* in the first half of the

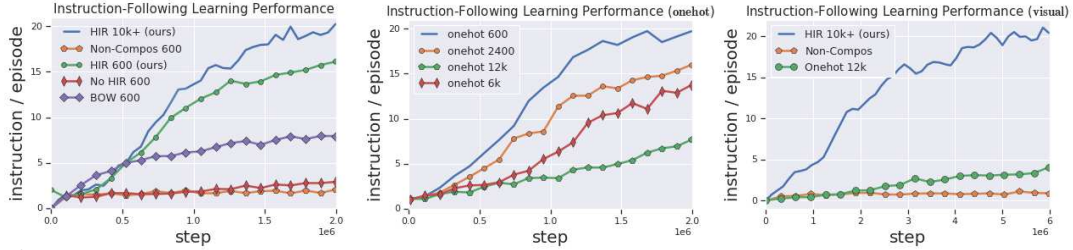

Figure 4: Results for low-level policies in terms of goals accomplished per episode over training steps for HIR. **Left**: HIR with different number of instructions and results with non-compositional representation and with no relabeling. **Middle**: Results for one-hot encoded representation with increasing number of instructions. Since the one-hot cannot leverage compositionality of the language, it suffers significantly as instruction sets grow, while HIR on sentences in fact learns even faster when instruction sets increase. **Right**: Performance of image-based low-level policy compared against one-hot and non-compositional instruction representations.

instructions. We emphasize that the agent has *never* seen the words red in the first part of the sentence in training; in other words, the task is *zero-shot* as the training set and the test set are disjoint (i.e. the distributions do not share support). From a pure statistical learning theoretical perspective, the agent should not do better than chance on such a test set. Remarkably, we observe that the agent generalizes better with language than with non-compositional representation (table 1). This suggests that the agent recognizes the compositional structure of the language, and achieves systematic generalization through such understanding.

|  | Standard train | Standard test | Standard gap | Systematic train | Systematic test | Systematic gap |
|---|---|---|---|---|---|---|
| **Language** | $21.50 \pm 2.28$ | $21.49 \pm 2.53$ | 0.001 | $20.09 \pm 2.46$ | $8.13 \pm 2.34$ | 0.596 |
| **Non-Compositional** | $6.26 \pm 1.18$ | $5.78 \pm 1.44$ | 0.077 | $7.54 \pm 1.14$ | $0.76 \pm 0.69$ | 0.899 |
| **Random** | $0.17 \pm 0.20$ | $0.21 \pm 0.17$ | - | $0.11 \pm 0.19$ | $0.18 \pm 0.22$ | - |

Table 1: Final performance of the low-level policy on different training and test instruction distributions (20 episodes). Language outperforms the non-compositional language representation in both absolute performance and relative generalization gap for every setting. *Gap* is equal to one minus the ratio of between mean test performance and mean train performance; this quantity can be interpreted as the generalization gap. For instructions with language representation, the generalization gap increases by approximately $59.5\%$ from standard generalization to zero-shot generalization while for non-compositional representation the generalization gap increases by $82.2\%$

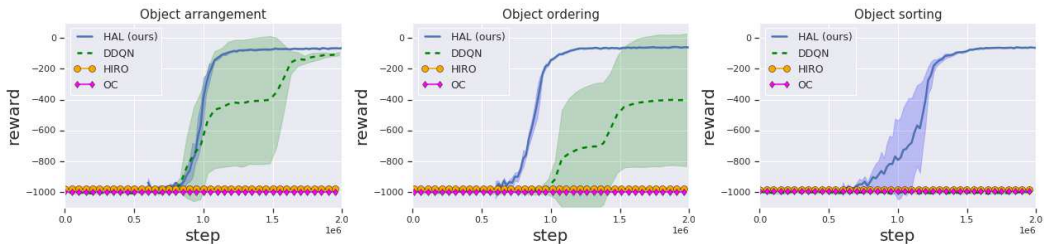

Figure 5: Results for high-level policy on tasks (a-c). Blue curves for HAL include the steps for training the low-level policy (a *single* low-level policy is used for all 3 tasks). In all settings, HAL demonstrates faster learning than DDQN. Means and standard deviations of 3 random seeds are plotted.

## 6.2 High-level policy

Now that we have analyzed the low-level policy performance, we next evaluate the full HAL algorithm. To answer **(4)**, we compare our framework in the state space against a non-hierarchical baseline DDQN and two representative hierarchical reinforcement learning frameworks HIRO [42] and Option-Critic (OC) [5] on the proposed high-level tasks with sparse rewards (Sec.5). We observe that neither HRL baselines are able to learn a reasonable policy while DDQN is able to solve only 2 of the 3 tasks. HAL is able to solve all 3 tasks consistently and with much lower variance and better asymptotic performance (Fig.5). Then we show that our framework successfully transfers to high-dimensional

observation (i.e. images) in all 3 tasks without loss of performance whereas even the non-hierarchical DDQN fails to make progress (Fig.6, left). Finally, we apply the method to 3 additional diverse tasks (Fig.6, middle). In these settings, we observed that the high-level policy has difficulty learning from pixels alone, likely due to the visual diversity and the simplified high-level policy parameterization. As such, the high-level policy for diverse setting receives state observation but the low-level policy uses the raw-pixel observation. For more details, please refer to Appendix B.3.

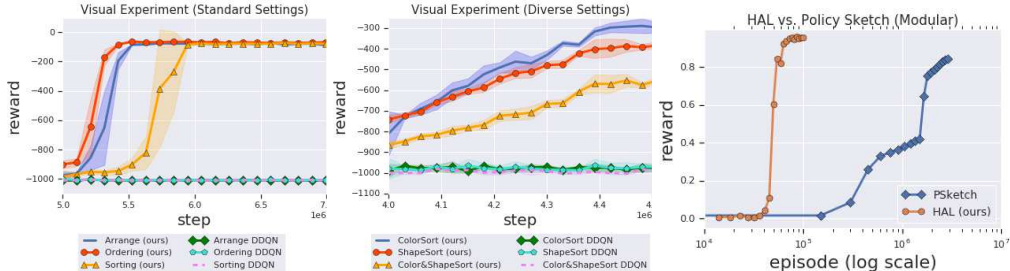

Figure 6: **Left**: Results for **vision-based** hierarchical RL. In all settings, HAL demonstrates faster and more stable learning while the baseline DDQN cannot learn a non-trivial policy. In this case, the vision-based low-level policy needs longer training time ($\sim 5 \times 10^6$ steps) so we start the x-axis there. Means and standard deviations of 3 seeds are plotted (Near 0 variance for DDQN). **Middle**: Results of HRL on the proposed 3 diverse tasks (d-e). In this case, the low-level policy used is trained on image observation for ($\sim 4 \times 10^6$ steps). 3 random seeds are plotted and training has not converged. **Right**: HAL vs policy sketches on the Crafting Environment. HAL is significantly more sample efficient since it is off-policy and uses relabeling among all modules.

## 6.3 Crafting Environment

To show the generality of the proposed framework, we apply our method to the **Crafting** environments introduced by Andreas et al. [2] (Fig. 6, right). We apply HAL to this environment by training *separate* policy networks for each module since there are fewer than 20 modules in the environment. These low-level policies receive binary rewards (i.e. one-bit supervision analogous to completing an instruction), and are trained jointly with HIR. Another high-level policy that picks which module to execute for a fixed 5 steps and is trained with regular DDQN. Note that our method uses a *different* form of supervision compared to policy sketch since we provide binary reward for the low-level policy – such supervision can sometimes be easier to specify than the entire sketch.

## 7 Discussion

We demonstrate that language abstractions can serve as an efficient, flexible, and human-interpretable representation for solving a variety of long-horizon control problems in HRL framework. Through re-labeling and the inherent compositionality of language, we show that low-level, language-conditioned policies can be trained efficiently without engineered reward shaping and with large numbers of instructions, while exhibiting strong generalizations. Our framework HAL can thereby leverage these policies to solve a range of difficult sparse-reward manipulation tasks with greater success and sample efficiency than training without language abstractions.

While our method demonstrates promising results, one limitation is that the current method relies on instructions provided by a language supervisor which has access to the instructions that describe a scene. The language supervisor can, in principle, be replaced with an image-captioning model and question-answering model such that it can be deployed on real image observations for robotic control tasks, an exciting direction for future work. Another limitation is that the instruction set used is specific to our problem domain, providing a substantial amount of pre-defined structure to the agent. It remains an open question on how to enable an agent to follow a much more diverse instruction set, that is not specific to any particular domain, or learn compositional abstractions without the supervision of language. Our experiments suggest that both would likely yield an HRL method that requires minimal domain-specific supervision, while yielding significant empirical gains over existing domain-agnostic works, indicating an promising direction for future research. Overall, we believe this work represents a step towards RL agents that can effectively reason using compositional language to perform complex tasks, and hope that our empirical analysis will inspire more research in compositionality at the intersection of language and reinforcement learning.

**Acknowledgments**

We would like to thank anonymous reviewers for the valuable feedback. We would also like to thank Jacob Andreas, Justin Fu, Sergio Guadarrama, Ofir Nachum, Vikash Kumar, Allan Zhou, Archit Sharma, and other colleagues at Google Research for helpful discussion and feedback on the draft of this work.

## Footnotes

\*Work done as a part of the Goolge AI Residency program

[2]Code and videos of the environment, and experiments are at https://sites.google.com/view/hal-demo

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
