[Supplementary Material]

# A Environment, Model Architectures, and Implementation

In this section, we describe the environment and tasks we designed, and various implementation details such as the architectures of the policy networks.

## A.1 Environment

We want an environment where we can evaluate the agent's ability to learn policies for long-horizon tasks that have compositional structure. In robotics, manipulating and rearranging objects is a fundamental way through which robots interact with the environment, which is often cluttered and unstructured. Humans also tend to decompose complex tasks into smaller sub-tasks in the environment (e.g. putting together a complicated model or write a piece of program). While previous works have studied the use of language in navigation domain, we aim to develop an environment where the agent can physically interact with and change the environment.

To that end, we designed a new environment for language and manipulation tasks in MuJoCo where the agents must interact with the objects in the scene. To succeed in this environment, the agents must be able to handle different number of objects with diverse visual and physical properties.

We will refer to all the elements in the environment collectively as the *world state*. The environment can contain up to 5 object. Each object is represented by $o_i$ that contains the *3d coordinate*, $\boldsymbol{p}_i$, of its center of mass, and a one-hot representation of its 4 properties: *color, shape, size*, and *material*. The environment keeps an internal relation graph $G_{adj}$ for all the objects currently in the scene. The relation graph is stored as an adjacency list whose $i^{th}$ entry is a nested array storing $o_i$'s neighbors in 4 cardinal directions *left, right, front* and *behind*. The criterion for $o_j$ to be the neighbor of $o_i$ in certain direction is if $||\boldsymbol{p}_j - \boldsymbol{p}_i|| \leq r_{\max}$ and the angle between $\boldsymbol{p}_j - \boldsymbol{p}_i$ and the cardinal vector of that vector is smaller than $\beta_{\max}$. After every interaction between the agent and the environment, $o_i$ and the relation graph are updated to reflect the current world state.

The agent takes the form of a point mass that can teleport around, which is a mild assumption for standard robotic arms (Other agents are possible as well). Before each interaction, the environment stores a set of language statements that are not satisfied by the current world state. These statements are re-evaluated after the interaction. The statements whose values change to True during the interaction can be used as the goals or instructions for relabeling the trajectories (cf. pre and post conditions used in classical AI planning ). Assuming the low-level policy only follows a single instruction at any given instant, the reward for every transition is 1 if the goal is achieved and 0 otherwise. The action space we use in this work consists of a point mass agent pushing one object in 1 of the 8 cardinal directions for a fixed number of frames, so the discrete action space has size $8k_t$, where $k_t \leq 5$ is the number of objects.

## A.2 Tasks

The high-level policy's reward function can be tailored towards the task of interests, where we propose three difficult benchmark tasks with extremely sparse rewards.

### A.2.1 Five Object Settings (Standard)

In this setting, we have a fixed set of 5 spheres of different colors *cyan, purple, green, blue, red*.

The first task we consider is **object arrangement**. We sample a random set of statements that can be simultaneously satisfied and, at every time step, the agent receives a reward of -10.0 if at least 1 statement is not satisfied satisfied and 0.0 only if all statements are satisfied. At the beginning of every episode, we reset the environment until none of the statements is satisfied. The exact arrangement constraints are: **(1)** red ball to the right of purple ball; **(2)** green ball to the right of red ball; **(3)** green ball to the right of cyan ball; **(4)** purple ball to the left of cyan ball; **(5)** cyan ball to the right of purple ball; **(6)** red ball in front of blue ball; **(7)** red ball to the left of green ball; **(8)** green ball in front of blue ball; **(9)** purple ball to the left of cyan ball; **(10)** blue ball behind the red ball

The second task is **object ordering**. An example of such a task is *"arrange the objects so that their colors range from red to blue in the horizontal direction, and keep the objects close vertically".* In this case, the configuration can be specified with 4 pair-wise constraint between the objects. We reset the environment until at most 1 pair-wise constraint is satisfied involving the x-coordinate and the

572 y-coordinate. At every time step, the agent receives a reward of -10.0 if at least 1 statement is not
573 satisfied satisfied and 0.0 only if all statements are satisfied. The ordering of color is: *cyan, purple,*
574 *green, blue, red* from left to right.

575 The third task is **object sorting**. In this task, the agent needs to sort 4 object around a central object;
576 further, the 4 objects cannot be too far away from the central object. Once again, the agent receives
577 a reward of -10.0 if at least 1 constraint is violated, and 0.0 only if all constraints are satisfied and
578 environment is reset until at most 1 constraint is satisfied. Images of end goal for each high-level
579 tasks are show in Figure 3.

### A.2.2 Diverse Object Settings

581 Here, instead of 5 fixed objects, we introduce 3 different shapes *cube, sphere* and *cylinder* in
582 combinations with 5 colors. Both colors and shapes can repeat but the same combination of color and
583 shape does not repeat. In this setting, there are $\binom{15}{5} = 3003$ possible object configurations. In this
584 setting, we define the color hierarchy to be *red, green, blue, cyan, purple* from left to right and the
585 shape hierarchy to be *sphere, cube, cylinder* from left to right. Sample goal states of each task are
586 shown in 3.

587 The first task is **color ordering** where the agent needs to manipulate the objects such that their colors
588 are in ascending order.

589 The second task is **shape ordering** where the agent needs to manipulate the object such that their
590 shapes are in ascending order.

591 Finally, the last task is **color & shape ordering** where the agent needs to manipulate the object such
592 that the color needs to be in ascending order, and within each color group the shapes are also in
593 ascending order.

594 Like in the fixed object setting, the agent only receives 0 reward when the objects are completely
595 ordered; otherwise, the reward is always -10.

### A.3 Implementation details

597 **Language supervisor**. In this work, each language statement generated by the environment is
598 associated with a *functional program* that can be executed on the environment's relation graph to
599 yield an answer that reflects the value of that statement on the current scene. The functional programs
600 are built from simple elementary operation such as querying the property of objects in the scene, but
601 they can represent a wide range of statements of different nature and can be efficiently executed on
602 the relation graph. This scheme for generating language statements is reminiscent of the CLEVR
603 dataset [23] whose code we drew on and modified for our use case. Note that a language statement
604 that can be evaluated is equivalent to a *question*, and the instructions we use also take the form of
605 questions. For simplicity and computational efficiency, we use a smaller subset of question family
606 defined in CLEVR that only involves pair-wise relationships (*one-hop*) between the objects. We plan
607 to scale up to full and beyond CLEVR scale in future works.

608 **State based low-level policy.** When we have access to the ground truth states of the objects in the
609 scene, we use an object-centric representation of states by assuming $s_t = \{o_i\}_{i=1}^{k_t}$, where $o_i \in \mathbb{R}^{d_o}$
610 is the state representation of object $i$, and $k_t$ is the number of objects (which can change over time).
611 We also assume $a_t = \{\alpha_i\}_{i=1}^{k_t}$, where each $\alpha_i \in \mathbb{R}^{d_\alpha}$ acts on individual the object $o_i$.

612 We implement a specialized universal value function approximator [46] for this case. To handle a
613 variable number of relations between the different objects, and their changing properties, we built a
614 goal-conditioned self attention policy network. Given a set of $k$ object $\{o_i\}_{i=1}^{k}$, we first create pair-
615 wise concatenation of the objects, $\mathcal{O} = \{o_i \| o_j\}_{j=1,i=1}^{k,k}$. Then we transform every pair-wise vectors
616 with a single neural network $f_1$ into $\mathcal{Z} = \{f_1(o_i \| o_j)\}_{j=1,i=1}^{k,k}$. A recurrent neural network with GRU
617 [9], $f_2$, embeds the instruction $g$ into a real valued vector $\widetilde{g} = f_2(g)$. We use the embedding to attend
618 over every pair of object to compute weights $\{w_i = \langle \widetilde{g}, z_i \rangle | z_i \in \mathcal{Z}\}$. We then compute a weighted
619 combination of all $p_i$ where the weights are equal to the softmax weights $\exp(w_i)/\sum_{j=1}^{k^2} \exp(w_j)$.
620 This combination transforms the elements of $\mathcal{Z}$ are combined into a single vector $\bar{z}$ of fixed size.
621 Each $o_i$ is concatenated with $\widetilde{g}$ and $\bar{z}$ into $o_i' = (o_i \| \widetilde{g} \| \bar{z})$. Then each $o_i'$ is transformed with the
622 another neural network $f_3$ whose output is of dimension $d_\alpha$. The final output $Q = \{f_3(o_i \| \widetilde{g} \| \bar{z})\}_{i=1}^{k}$

is in $\mathbb{R}^{k \times d_\alpha}$ which represents all state-action values of the state. Illustration of the architecture is shown in Figure 7.

Figure 7: Computation graph of the state-based low level policy.

**Image based low-level policy.** In reality, we often do not have access to the state representation of the scene. For many application, a natural alternative is images. We assume $s_t \in [0, 1]^{W \times H \times C}$ is the available image representation of the scene (in all experiemnts, W=64, H=64, C=3). Further, we need to adopt a more general action space since we no longer have access to the state representation (e.g. coordinates of the location). To that end, we discretize the 2D space in to $10 \times 10$ grids and an action involves picking an starting location out of the 100 available grid and a direction out of the the 8 cardinal direction to push. This induces an 800 dimensional discrete action space.

It is well-known that reinforcement learning from raw image observation is difficult, and even off-policy methods require millions of interaction on Atari games where the discrete action space is small. Increasing the action space would understandably make the already difficult exploration problem harder. A potential remedy is found in the fact that these high dimensional action space can often be factorized into semantically meaningful groups (e.g. the pushing task can be break down into discrete bins of the $x$ and $y$ axes as well as a pushing direction). Previous works attempted to leverage this observation by using auto-regressive models or assuming conditional independence between the groups [32, 55]. We offer a new approach that aims to make the least assumptions. Following the group assumption, we assume there exists $m = 3$ groups and each group consists of $k_m$ discrete *sub-actions* (i.e. $\mathcal{A}_m = \{a_m^{(1)}, a_m^{(2)}, \ldots a_m^{(k_m)}\}$). Following this definition, we can build a bijective look-up map $\zeta$ between $\mathcal{A}$ to tuples of sub-actions:

$$\mathcal{B} = \prod_{n=1}^{m} \{1, 2, \ldots, k_n\} \tag{2}$$

$$\mathcal{A} \xrightarrow{\zeta} \{(a_1^{(i_1)}, \ldots, a_m^{(i_m)}) \mid \forall (i_1, \ldots, i_m) \in \mathcal{B}\} \tag{3}$$

We overload the notion $a_m^{(i)}(s)$ to the *action feature* of $a_m^{(i)}$ conditioned on the state and goal and $\zeta(a, s)$ to be a tuple of the corresponding action features. Then the value function each action can be represented as:

$$Q(s, a) = f_\psi(\zeta(a, s)) \tag{4}$$

where $f_\psi$ is a single neural network parameterized by $\psi$ that is shared between all $a$. This model does not require picking an order like the auto-regressive model and does not assume conditional independence between the groups. Most importantly, the number of parameter scales sublinearly with the dimension of the actions. The trade-off of this model is that it can be memory-expensive to model the full joint distribution of actions at the same time; however, we found that this model performs empirically well for the pushing tasks considered in this work. We will refer to this operation as *Tensor Concatenation*.

The overall architecture for the UVFA is as follows: the image input is fed through 3 convolution layers with kernel size $\{8, 5, 3\}$, stride $\{2, 2, 1\}$, and channel $\{46, 128, 64\}$; each convlution block is FiLM'd [39] with the instruction embedding. Then the activation is flattened spatially to $256 \times 64$

and projected to $28 \times 64$. This is further split into 3 action group of sizes $\{10 \times 64, 10 \times 64, 8 \times 64\}$ and fed through tensor concatenation. $f_\psi$ is parameterized with 2-layer MLP with 512 hidden units at each layer and output dimension of 1 which is the Q-value.

Figure 8: Computation graph of the vision-based low level policy.

Both policy networks are trained with HIR. Training mini-batches are uniformly sampled from the replay buffer. Each episode lasts for 100 steps. When the current instruction is accomplished, a new one that is currently not satisfied will be sampled. To accelerate the initial training and increase the diversity of instructions, we put a 10 step time limit on each instruction, so the policy does not get stuck if it is unable to finish the current instruction.

**High-level policy.** For simplicity, we use a Double DQN [58] to train the high-level policy. We uses an instruction set that consists of 80 instructions ($|\mathcal{I}| = 80$ for standard and $|\mathcal{I}| = 240$ for diverse) that can sufficiently cover all relationships between the objects. We roll out the low-level policy for 5 steps for every high-level instruction ($T' = 5$). Training mini batches are uniformly sampled from the replay buffer. The state-based high-level policy uses the same observation space as the low-level policy; the image-based high-level policy uses extracted visual features from the low-level policy and then extract salient spatial points with spatial softmax [29]. The convolutional weights are frozen during training. This design choice makes natural sense since humans also use a single visual cortex to process all initial visual signals, but training from scratch is certain possible, if not ideal, should computation budget not matter. For the diverse visual tasks, we found that using the convolutional features with spatial softmax could not yield a good representation for the downstream tasks. Due tot time constraints, the experiments shown for the diverse high-level tasks use the ground truth state, namely position and one-hot encoded colors and shapes for the high-level policy; however, the low-level policy only has access to the image. We believe a learned convolutional layer would solve this problem. Finally, we note that the high-level policy picks each sentence independent and therefore does not leverage the structure of language. While generating language is generally hard, a generative model would correspond to *thought* more. We think this is an extremely important and exciting direction for future work.

# B Algorithms

In this section we elaborate on our proposed algorithm and lay out finer details.

## B.1 Overall algorithm

The overall hierarchical algorithm is as follows:

## B.2 Training low-level policy

For both state-based and vision-based experiments, we use DDQN as $\mathcal{A}_l$. For the state-based experiments, the agent receives binary reward based on whether the action taken completes the instruction; for the vision-based experiments, we found it instrumental to add a object movements bonus, i.e. if the agents change the position of the objects by some minimum threshold, the agetn receives a 0.25 reward. This alleviates the exploration problem in high dimensional action space

---

**Algorithm 1** Overall Hierarchical Training

---

1: **Inputs:** Low level RL algorithm $\mathcal{A}_l$; High level RL algorithm $\mathcal{A}_h$; Environment $\mathcal{E}$; other relevant inputs of Algorithms 2 and 3
2: $\pi_l(\boldsymbol{a}|\boldsymbol{s}, \boldsymbol{g}) \leftarrow$ low level policy trained with $\mathcal{A}_l$ and other appropriate inputs (Algorithm 2)
3: $\pi_h(\boldsymbol{g}|\boldsymbol{s}) \leftarrow$ high level policy trained with $\mathcal{A}_h$, $\pi_l(\boldsymbol{a}|\boldsymbol{s}, \boldsymbol{g})$ and other appropriate inputs (Algorithm 3)
4: **return** $\pi_l(\boldsymbol{a}|\boldsymbol{s}, \boldsymbol{g})$ and $\pi_h(\boldsymbol{g}|\boldsymbol{s})$

---

---

**Algorithm 2** RL with hindsight instruction relabeling (HIR)

---

1: **Inputs:** off-policy RL algorithm $\mathcal{A}_l$; instruction relabeling strategy $\mathcal{S}$; language supervisor $\Omega$; Environment $\mathcal{E}$; number of relabeled future $K$
2: **Initialize** replay buffer $\mathbb{B}$ and $\pi_l(\boldsymbol{a}|\boldsymbol{s}, \boldsymbol{g})$
3: **for** episode $i = 1$ **to** $M$ **do**
4:      $\boldsymbol{s}_0 \leftarrow$ reset $\mathcal{E}$
5:      $\boldsymbol{g} \sim \mathcal{U}(\{\boldsymbol{g} \in \Omega(\boldsymbol{s}_0) \mid \Psi(\boldsymbol{s}_0, \boldsymbol{g}) = 0\})$
6:      $\tau \leftarrow [\,]$
7:      **for** step $t = 0$ **to** $T$ **do**
8:          $\mathbb{U}_t \leftarrow \{\boldsymbol{g} \in \Omega(\boldsymbol{s}_t) \mid \Psi(\boldsymbol{s}_t, \boldsymbol{g}) = 0\}$
9:          $\boldsymbol{a}_t \sim \pi_{\mathcal{A}}(\boldsymbol{a}|\boldsymbol{s}_t, \boldsymbol{g})$
10:         $\boldsymbol{s}_{t+1} \leftarrow$ Take action $\boldsymbol{a}_t$ from $\boldsymbol{s}_t$
11:         $r_t \leftarrow \Psi(\boldsymbol{s}_{t+1}, \boldsymbol{g})$
12:         $\mathbb{V}_t \leftarrow \mathbb{U}_t \setminus \{\boldsymbol{g} \in \Omega(\boldsymbol{s}_{t+1}) \mid \Psi(\boldsymbol{s}_{t+1}, \boldsymbol{g}) = 0\}$
13:         add $(\boldsymbol{s}_t, \boldsymbol{a}_t, \boldsymbol{g}, r_t, \boldsymbol{s}_{t+1}, \boldsymbol{a}_{t+1}, \mathbb{V}_t)$ to $\tau$
14:         **if** $r_t = 1$ **then**
15:            $\boldsymbol{g} \sim \mathcal{U}(\{\boldsymbol{g} \in \Omega(\boldsymbol{s}_{t+1}) \mid \Psi(\boldsymbol{s}_{t+1}, \boldsymbol{g}) = 0\})$
16:         **end if**
17:      **end for**
18:      **for** step $t = 0$ **to** $T$ **do**
19:         $\mathbb{B} \leftarrow \mathbb{B} \cup \{(\boldsymbol{s}_t, \boldsymbol{a}_t, \boldsymbol{g}, r_t, \boldsymbol{s}_{t+1}, \boldsymbol{a}_{t+1})\}$
20:         **for** $\boldsymbol{g}' \in \mathbb{V}_t$ **do**
21:            $\mathbb{B} \leftarrow \mathbb{B} \cup \{(\boldsymbol{s}_t, \boldsymbol{a}_t, \boldsymbol{g}', 1, \boldsymbol{s}_{t+1}, \boldsymbol{a}_{t+1})\}$
22:         **end for**
23:         $\mathbb{W} \leftarrow \mathcal{S}(\tau, t, K)^{\dagger}$
24:         **for** $(\boldsymbol{g}', r') \in \mathbb{W}$ **do**
25:            $\mathbb{B} \leftarrow \mathbb{B} \cup \{\boldsymbol{s}_t, \boldsymbol{a}_t, \boldsymbol{g}', r', \boldsymbol{s}_{t+1}, \boldsymbol{a}_{t+1}\}$
26:         **end for**
27:      **end for**
28:      Update $\pi_l(\boldsymbol{a}|\boldsymbol{s}, \boldsymbol{g})$ with $\mathcal{A}_l$ using minibtach from $\mathbb{B}$
29: **end for**
30: **return** $\pi_l(\boldsymbol{a}|\boldsymbol{s}, \boldsymbol{g})$
31: $\dagger$ Details in Appendix B.4. $\mathcal{U}(\cdot)$ denotes uniform sampling from the given set.

---

($\mathbb{R}^{800}$), but the algorithm is capable of learning without the exploration bonus. (Algorithm 2). We adopt the similar setting as HER where unit of time consists of epochs, cycles and episode. Each cycle consists of 50 episode and each episode consists of 100 steps. While we set the number of epoch to 50, in practice we never actually reach there. We adopt an epsilon greedy exploration strategy where every cycle we decrease the exploration by a factor of 0.993, starting from 1 but at the beginning we 10 cylces to populate the buffer. The minimum epsilon is 0.1. We use $\gamma = 0.9$ and replay buffer of size 2e6. The target network we use is a 0.95 moving average of the model parameters, updated at the beginning of every cycle. Every episode, we update the network 100 steps with the Adam Optimizer with minibatch randomly sampled from the replay buffer.

## B.3 Training high-level policy

$\mathcal{A}_h$ is also a DDQN. One single set of hyperparameter is used for standard experiments and another for the diverse experiments. DDQN is trained for 2e6 steps, uses uniform replay buffer of size 1e5,

linearly anneal epsilon from 1 to 0.05 in 3e5 steps, Adam and batch size 256. $T' = 5$ for all all our experiments, but the experience the network sees is equivalent of 1 step. For the diverse settings, do to time constraint, we use priority replay buffer of size 1e6 for all diverse experiment including the DDQN baselines. (Algorithm 3) We use the Adam Optimizer [26] with initial learning rate of 0.0001. Discounte factor $\gamma = 0.9$. Learning starts after 100 episode each of which lasts for 100 steps / 100 high-level actions.

---

**Algorithm 3** Training high-level policy

---

1: **Inputs:** Any RL algorithm $\mathcal{A}_h$; reward function $R : \mathcal{S} \rightarrow [r_{\min}, r_{\max}]^*$; instruction set $\mathcal{I}$; instruction encoder $\phi$; low-level policy $\pi_l(\boldsymbol{a}|\boldsymbol{s}, \boldsymbol{g})$
2: **Initialize** $\mathcal{A}$
3: **for** episode $i = 1$ **to** $M$ **do**
4:   $\boldsymbol{s}_0 \leftarrow$ reset $\mathcal{E}$
5:   **for** step $t = 0$ **to** $T$ **do**
6:     $\boldsymbol{g} \leftarrow$ Sample from $\mathcal{I}$ using $\pi_h(\boldsymbol{g}|\boldsymbol{s}_t)$
7:     $\boldsymbol{s}' \leftarrow \boldsymbol{s}_t$
8:     **for** substep $t' = 1$ **to** $T'$ **do**
9:       $\boldsymbol{a}' \sim \pi_l(\boldsymbol{a}|\boldsymbol{s}', \boldsymbol{g})$
10:       $\boldsymbol{s}' \leftarrow$ Take action $\boldsymbol{a}'$ from $\boldsymbol{s}'$
11:     **end for**
12:     $\boldsymbol{s}_{t+1} \leftarrow \boldsymbol{s}'$
13:     Store experience
14:   **end for**
15:   Update $\mathcal{A}_h$ accordingly with experience collected
16: **end for**
17: *Here we assume the reward is only based on the new state for simplicity

---

### B.4   Relabeling Strategy

HER [2] demonstrated that the relabeling strategy for trajectories can have significant impacts on the performance of the policy. The most successful relabeling strategy is the "k-future" strategy where the goal state and the reward are relabeled with $k$ states in the trajectories that are reached *after* the current time step and the reward is discounted based on the discount factor $\gamma$ and how far away the current state is from the future state in $\ell_2$ distance. We modify this strategy for relabeling a language conditioned policy. One challenge with language instruction is that the notion of distance is not well defined as the instruction is under-determined and only captures a part of the information about the actual state. As such, conventional metrics for describing distance between sequences of tokens (e.g. edit distance) do not actually capture the information we are interested in. Instead, we adopt a more "greedy" approach to relabeling by putting more focus on 1-step transition where the instruction is actually fulfilled. Namely, we store all transition tuples in $\mathbb{V}_t$ to the replay buffer $\mathbb{B}$ (Algorithm 2). For future relabeling, we simply use the reward discounted by time steps into the future to relabel the trajectory. While the discounted reward does not usually capture the "optimal" or true discounted reward, we found it to provide sufficient learning signal. Detailed steps are shown below (Algorithm 4). In our experiments, we use $K = 4$.

In additon to relabeling, if an object is moved (using 800 dimensional action space), we add to replay buffer a transition where the instruction is the name of the object such as *"large rubber red ball"* and the reward is $1.0$. We found this helps the agent to learn the concept of objects. We refer to this operation as *Unary Relabeling*.

## C   Experimental Details

### C.1   One-hot encoded representation

We assign each instruction a varying number of bins in the one-hot vector. Concretely, we give each instruction of the 600 instruction 1, 4, 10, and 20 bins in the one-hot vector, which means the effective size of the one-hot vector is 600, 2400, 6000 and 12000. When sampling goals, each goal is uniformly dropped into one of its corresponding bins.

**Algorithm 4** Future Instruction Relabeling Strategy ($\mathbb{S}$)

---

1: **Inputs:** Trajectory $\tau$; current time step $t$; number of relabeled future $K$
2: $\Delta \leftarrow [\,]$
3: $\texttt{count} \leftarrow 0$
4: **while** $\texttt{count} < K$ **do**
5:    $\texttt{future} \sim \mathcal{U}(\{t+1, \ldots, |\tau|\})$
6:    $(s, a, g, r, s', a', \mathbb{V}) \leftarrow \tau[\texttt{future}]$
7:    **if** $|\mathbb{V}| > 0$ **then**
8:       $g' \sim \mathcal{U}(\mathbb{V})$
9:       $r' \leftarrow r \cdot \gamma^{\texttt{future}-t}$
10:       Store $(g', r')$ in $\Delta$
11:       $\texttt{count} \leftarrow \texttt{count} + 1$
12:    **end if**
13: **end while**
14: **return** $\Delta$

---

## C.2   Non-compositional representation

To faithfully evaluate the importance of compositionality, we want a representation that carries the identical information as the language instruction but without the explicit compositional property (but perhaps still to some degree compositional). To this end, we use a Seq2Seq [53] autoencoder with 64 hidden units to compress the 600 instructions into real-valued continuous vectors. The original tokens are fully recovered which indicates that the compression is *lossless* and the latent's information content is the same as the original instruction. This embedding is used in place of the GRU instruction embedding. We also observed that adding regularization to the autoencoder decreases the performance of the resulting representation. For example, decreasing the bottleneck size leads to worse performance, so does adding dropout. Figure 4 uses an autoencoder with dropout of 0.5 while 1 uses one with no dropout. As you can see, the performance without dropout is better than the one with. We hypothesis adding regularization decreases the compositionality of the representation.

## C.3   Non-hierarchical baseline

We use Double DQN implementation from OpenAI baselines (https://github.com/openai/baselines/tree/master/baselines/deepq). We use a 2 layer MLP with 512 hidden units at each layer with the respective action dimension as the output as the policy.

## C.4   HRL baselines

In general, we note that it is difficult to compare different HRL algorithms in an apple-to-apple manner. HIRO assumes a continuous goal space so we modified the goal to be an $\mathbb{R}^{10}$ vector representing the locations of each object rather than using language. In this regime, we observed HIRO was unable to make good progress. We hypothesize that the highly sparse reward might be the culprit. It is also worth noting that HIRO uses a goal space in $\mathbb{R}^2$ for navigation (which is by itself a choice of abstraction because the actual agent state space is much higher) while ours is of higher dimensionality. The Tensorflow implementation of HIRO we use can be found at https://github.com/tensorflow/models/tree/master/research/efficient-hrl. (This is the implementation from the original author).

Option-critic aims to learn everything in a complete end-to-end manner which means it does not use the supervision from language, which makes it perhaps not as surprising that the sparse tasks do not provide sufficient signal for OC. In other words, while our method enjoys the benefit of a flexible but fixed abstraction while OC needs to learn such abstraction. We tried 8, 16, and 32 options for OC but our method has much more sub-policies due to the combinatorial nature of language. The OC implementation in Tensorflow we used can be found at https://github.com/yadrimz/option-critic.

 **C.5 Hardware Specs and Training time**

All our experiments are performed on a single Nvidia Tesla V100. We are unable to verify the specs of the virtual CPU. The low-level policy for state-based observation takes about 2 days to train and, for image-based observation, 6 days. The high-level policies for the state-based observation takes about 2 days to train and 3 days for image-based observations (wall-clock time). The implementations are not deliberately optimized for performance (major bottleneck is actually the language supervisor) so it is very likely the time could be dramatically shortened.

# D More Experimental Results

## D.1 Low-level policy for diverse environment

Figure 9 shows the training instruction per episode on the diverse environment. We see that the performance is worse than fixed number of objects with the same amount of experience. This is perhaps not surprising considering the visual tasks are much more diverse and hence more challenging.

Figure 9: Results training the low-level policy on the diverse environment.

## D.2 Why is the proposed environment difficult?

While DDQN worked on 2 cases in the state-based environemnt, it is unable to solve any of the problems in visual domain. We hypothesize that the pixel observations and increase in action space ($20\times$ increase) makes the exploration difficult for DDQN. The difficulty of the tasks–in particular the 3 standard tasks–is reflected in the fact that the reward from non-hierarchical random action is stably 0 with small variance, meaning that under the sparse reward setting the agent rarely visits the goal state. On the other hand, the random exploration reward is much higher for our method as the exploration in the space of language is *structured*.