[Reviews · NeurIPS 2019]

Reviewer 1



I believe the proposed method, HAL (Hierarchical Abstraction with Language), is an interesting approach for HRL. The authors adapt Hindsight Experience Replay for instructions (called Hindsight Instruction Relabelling). I have some concerns about the experimental setup and empirical evaluation of the proposed method: - The motivation behind introducing a new environment is unclear. There are a lot of similar existing environments such as crafting environment used by [1], compositional and relational navigation environment in [2]. Introducing a new environment (unless its necessary) hinders proper comparison and benchmarking. It seems to me that the environment was specifically designed to highlight the strengths of the proposed method. One of the most important motivations behind studying HRL methods is solving complex sparse reward tasks. I would have liked to see the proposed method applied to some of the most popular sparse rewards tasks such as Montezuma's Revenge, as it helps in gauging the significance of the proposed method as compared to several published methods evaluated on these tasks. If the proposed method can not be applied to standard HRL tasks, then it is a limitation of the method which should be discussed. - I believe the proposed method is similar to [1] and [3]. The authors should position their work with respect to [1] and [3] which would also serve as better baselines in my opinion. - Two HRL methods used as baselines in the experiments completely fail in the new environment proposed by the authors. Some explanation behind this result would be helpful. - The authors state that the high-level policy always (even in the diverse setting) uses ground-truth state as input, namely position and one-hot encoded colors and shapes. I think the ground-truth state consists of compositional features which make it very easy for the high-level policy to learn to output instructions, while the baselines can not leverage this. I believe this is an unfair comparison and allows raises concerns about the effectiveness of high-level policy with high-dimensional state space, especially because it is not tested on any standard environment. - The appendix is full of typos. For example, line 675, "Due tot", line 705 "do to time constraint". Section B.1, which is referenced several times in the main paper, seems to be incomplete. [1] Jacob Andreas, Dan Klein, and Sergey Levine. 2017. Modular multitask reinforcement learning with policy sketches. ICML-17 [2] Yu, H., Zhang, H., & Xu, W. (2018). Interactive grounded language acquisition and generalization in a 2d world. ICLR-18 [3] Oh, Junhyuk, et al. "Zero-shot task generalization with multi-task deep reinforcement learning." ICML-2017. ---- Updated after author response: After reading the author response and other reviews, I maintain my rating. This is due to the following reasons: 1) It seems to me that HAL is specifically designed for the proposed environment and not a general HRL method. The author response confirms that the proposed method is not general enough to be applied to any environment ("environments like Montezuma’s don’t have labeled data or infrastructures for complex language captioning"), however the introduction claims that HAL is a general HRL method, the first contribution stated is "a framework for using language abstractions in HRL, with which we find that the structure and flexibility of language enables agents to solve challenging long-horizon control problems". Specifically, the environment needs to provide whether each language statement is satisfied or not by the current world state. I believe the crafting environment implemented by the authors also provides this information. This is a very strong assumption and severely limits the applicability of an HRL method. These limitations are not acknowledged in the submission. I suggest reframing the introduction to introduce the task/environment and the challenges associated with it, and propose a solution for the specific task. 2) Based on the author response, I believe most of the gains over the baselines are coming from Hindsight Instruction Relabelling (as the authors also mention "DDQN is able to solve only 2 of the 3 tasks, likely due to the sparse reward" in Section 6.2, and in the rebuttal authors say "HAL significantly outperforms policy sketch because it is off-policy and leverages hindsight relabeling"). In my opinion, HIR is an adaptation of HER in the proposed environment and not very original. 3) The above also raises concerns about fairness in comparison with other methods. HIR requires specific information from the environment about whether each language statement is satisfied or not by the current world state. This makes the comparison unfair because baselines do not leverage this information from the environment. 4) I also agree with Reviewer 3's concern about high-level policy only chooses from a fixed instruction subset and therefore does not learn or output anything compositional. The additional results provided in the author response are significantly different from the original submission and require additional details.

Reviewer 2



Originality and Significance: The idea is natural and intuitive. Now that the authors have shown this idea works, there's a direct avenue for incorporating ideas from other work (i.e., generalization in visual QA) to improving RL. The authors did a great job finding the right setting (a reasonably compositional one) to showcase language's promise in RL (highlighted by the systematic generalization results). I know that several others have been thinking about this idea in general for a while (using language as an abstraction in HRL) - for example, see concurrent/later work "Hierarchical Decision Making by Generating and Following Natural Language Instructions" (https://arxiv.org/pdf/1906.00744.pdf). Regardless, it is great to see this idea actually work in a pretty challenging / sparse reward RL setting. One drawback of the implemented agent is that the high-level policy treats each instruction distinctly, which takes away from some of the story of aiding RL by exploiting the compositionality in language. Decoding instructions in a compositional manner would be fit better with the authors' aim; full autoregressive decoding would be impressive (but challenging), but it would even just be interesting to factor the action space compositionally (e.g., first predict the instruction template, then predict the key nouns/adjectives in the template, or perhaps using hierarchical softmax to decode actions). Right now, as I understand, the low-level policy treats the goal input as compositional, but the high-level policy does not treat actions as compositional. Quality: The work is well-executed. Task-design and model-design decisions are simple, clear, and well-motivated. The instructions themselves could be more diverse; I would've been more interested in seeing the authors experiment on highly compositional/diverse instructions (i.e., at the level of language complexity/compositionality of CLEVR questions) on the state-based environment rather than experimenting with simpler language instructions from pixel-based observations (since the paper's focus is on language). Clarity: The writing was quite clear overall. In general, I felt like the paper made many distinct points about how language could be useful; it would've been helpful to frame the intro/discussion/paper as focusing on 1-2 of these (i.e., compositional generalization), as well as being concrete about how language can help. For example, it seems that compositional generalization through language is a relatively unique/strong aspect of this work, while using language instructions to specify vague goals is a property of instruction-following tasks in general, not specific to using language in HRL. I only understood around Page 7 (experiments) why the authors concretely expected language to help with generalization (when the authors describe the explicitly non-compositional approaches); even then, I would've liked more explanation for why policies did generalize compositionally, in contrast with the expectation (Page 7): "From a pure statistical learning theoretical perspective, the agent should not do better than chance on such a test set." Minor writing comments: * "Fortunately, while the the size" -> "Fortunately, while the size" * "arrnage 4 objects around an central object" -> "arrange 4 objects around a central object" * Figure 4b: Maybe order the keys in the figure by the number of instructions / the performance in the graph (more intuitive/easier to read). And/or spell out "12k" -> "12000" so faster to tell what's going on (initially I was confused reading the legend) * Figure 5: The legend is pretty small * The appendix has a few typos as well

Reviewer 3



Thanks to the authors for the response - these new experiments certainly are a step towards better demonstrating the role of compositionality in this work. However, these experiments need elaboration and further analysis, especially since the formulation of the high-level policy is a new one. This changes the paper quite a bit and I feel would necessitate another round of evaluation. --------- This paper proposes to use instructions in natural language as a way of specifying subgoals in a hierarchical RL framework. The authors first train a low-level policy to follow instructions (using dense rewards) and a high-level policy to generate/choose instructions (using sparse environmental rewards). The goal is to enable better generalization of HRL algorithms and to solve the sparse reward problem. I really like the idea of this paper - it is definitely novel and worth pursuing and the paper is written clearly. However, the execution is a bit lacking and the experiments do not clearly demonstrate the advantage of using compositional language, which is the main premise of the paper. Re: compositionality: While the paper's idea of using the compositional nature of language to improve HRL is definitely interesting, the experiments do not back up the claims. First, the high-level policy only chooses from a fixed instruction subset and therefore does not learn or output anything compositional. Second, the non-compositional baseline for the low-level policy doesn't make sense. Why should a lossless representation be non-compositional, especially with a sequence auto-encoder? For fair comparison with the language representation of subgoals, the auto-encoder representation should also be fed into a GRU and not used directly? Further, another baseline would be to simply have a bag-of-words representation for each instruction (note that this is different from having a one-hot representation for each instruction). From the current experiments, it is unclear that there is an advantage to using language as an intermediate representation (even though theoretically it makes perfect sense!). Other comments: 1. Some of the experimental details are not clear. Do you assume access to the ground truth state of the world for the HIR procedure? If so, this should be made clear as an assumption. 2. Are the number of actions used for the high-level policy the same for your method (80) vs the other HRL baselines? From C.4, it looks like the goal space is much larger for the baselines. 3. (minor) How well does the model generalize to real instructions (with potential noise)? 4. Relevant work: a) Speaker-follower models for vision-and-language navigation D Fried, R Hu, V Cirik, A Rohrbach… - Advances in Neural …, 2018 - papers.nips.cc b) Grounding language for transfer in deep reinforcement learning K Narasimhan, R Barzilay, T Jaakkola - Journal of Artificial Intelligence …, 2018 - jair.org c) Vision-based Navigation with Language-based Assistance via Imitation Learning with Indirect Intervention Khanh Nguyen, Debadeepta Dey, Chris Brockett, Bill Dolan d) Learning to Navigate Unseen Environments: Back Translation with Environmental Dropout Hao Tan, Licheng Yu, Mohit Bansal

[Author Response · NeurIPS 2019]

We thank all reviewers for the constructive comments. We first present new experimental results requested by the
reviewers, and then address reviewers' concern individually. All new results will be included in the revision.

**R1** : We added a comparison to **policy sketches** (Andreas et al. '17) on the **Crafting** environment from the same work.
HAL significantly outperforms policy sketch because it is off-policy and leverages hindsight relabeling, and also does
not require sketch supervision (see Fig. 1, left). This also shows HAL works well on an environment from prior work.

**R2** , **R3** : We found that naively training a high-level policy to output language directly works poorly, as policy opti-
mization and language generation destabilize one another. To get around this problem, we pre-trained an autoregressive
language model as the head of the policy, resulting in a **high-level policy that generates instructions directly**. This
approach can leverage compositionality and does not limit the number of instructions, but (with uniform replay buffer)
it currently performs comparably due to the challenges of optimizing the language model. (See Fig. 1, middle.)

**R3** : We compare to using a **bag-of-words** representation that ignores the **sequential nature** of instructions. Shown in
Figure 1, right, our approach significantly outperforms the bag of words.

Figure 1: **Left**: Crafting Environment. **Middle**: Language Model High-level Policy. **Right**: Language vs Bag-of-words.

**R1 Re:comparison**: We clarify that only the high-level policy in the more challenging diverse setting uses object state
while all other settings operate on RGB inputs. In all settings, HIRO and OC use ground truth state and perform poorly,
while HAL works well both with the same ground truth state, and in the non-diverse setting, with pixel observations.

**Re: similar methods**: We added a comparison to Andreas et al. (see top). Both suggested methods (Andreas et al. '17
and Oh et al. '17) require direct supervision on which instructions to perform for each high-level training task. Our
method does not require this kind of high-level task supervision. Hence, HAL is more directly comparable to HRL
algorithms like OC and HIRO, which we compare to in the paper.

**Re: environment** We evaluated HAL on the suggested environment from prior work (see top). As R2 pointed out, the
proposed environment exhibits significantly more compositionality than prior RL and language environments. To our
knowledge, environments like Montezuma's don't have labeled data or infrastructures for complex language captioning.

**Re: baseline failures** We used the official implementations of HIRO, and consulted the authors of HIRO to ensure that
the method is working as expected. For OC, all the options quickly start to terminate after 1 step, which is a known
failure mode of OC (e.g., see Nachum et al. '18). Please see C.4 for more analysis, to which we will add more.

**R2 Re: analysis** We hypothesize that language achieves superior zero-shot generalization because the language in the
training and test instructions obey the same *grammar* that specifies how concepts combine. We will discuss this further
in the revised version. Successful instructions can be found in the captions of the videos on the supplementary website.

**Re: failure modes** Some notable failure modes are the following. 1. Increasing visual diversity (e.g. texture, sizes)
poses a challenge, possibly due to the model capacity. 2. When the high-level policy outputs instructions that contain
non-existent objects, the low-level policy tends to fail to induce any changes, causing the high-level policy to keep
giving the same instruction (Diverse setting's videos have a few examples of this). We will add more detailed analysis.

**Re: diversity** The environment readily supports more instructions that will be available at release. At 1 million steps,
HIR achieves 3.9 instruction/episode (0.2 for random) on the combination of CLEVR's *1, 2 and 3 hop* questions.

**R3 Re: compositionality** We added a comparison to bag-of-words (see top). We clarify that the latent code of a
sequence auto-encoder is not a sequence, but a fixed-length continuous vector (Appendix C.2). This is a *latent variable*
model, as suggested. We agree this latent representation may not be completely non-compositional, but our experiments
showcase the benefit of language over the latent variables. We will make this more clear in the revision.

**Re: assumption** This work uses instructions generated by the CLEVR engine, which use ground-truth locations of the
objects. In principle, these instructions could also be generated by a learned captioning-like model. We will edit Sec. 1
and 3 to make this more clear.

**Re: high-level action space** OC uses a smaller number of discrete options, as it does not work with large number of
options (e.g. 80). HIRO specifies continuous goals in coordinates of the objects, and hence has the same dimensionality
(10) which is very sensible for continuous control RL tasks (e.g. Ant has 7 DOF and Humanoid has 17 DOF).

**Re: related work** Thank you for the suggested references! We will cite and discuss the referenced papers.

[Meta-Review · NeurIPS 2019]

The additional experiments presented in the abstract addressed many of the reviewers concerns; however, there was some doubt that these changes will be successfully incorporated into a camera ready. These additions (especially the use of a full language model in the policy and the crafting world results) would significantly strengthen the paper and I strongly urge the authors to follow through on their rebuttal commitment of integrating these results in future revisions. There are also concerns that the approach is highly specialized for the environment and is limited by its need for automatic goal language prediction / verification to perform HIL. Given the content of the paper already, this might be better left to future work.